# A Commentary on the Healthy Community Stores Case Study Project: Implications for Retailers, Policy, and Future Research

**DOI:** 10.3390/ijerph19148824

**Published:** 2022-07-20

**Authors:** Samantha M. Sundermeir, Megan R. Winkler, Sara John, Uriyoán Colón-Ramos, Ravneet Kaur, Ashley Hickson, Rachael D. Dombrowski, Alex B. Hill, Bree Bode, Julia DeAngelo, Joel Gittelsohn

**Affiliations:** 1Department of International Health, Bloomberg School of Public Health, Johns Hopkins University, Baltimore, MD 21205, USA; jgittel1@jhu.edu; 2Department of Behavioral, Social and Health Education Sciences, Rollins School of Public Health, Emory University, Atlanta, GA 30322, USA; megan.winkler@emory.edu; 3Center for Science in the Public Interest, Washington, DC 20005, USA; sjohn@cspinet.org (S.J.); ahickson@cspinet.org (A.H.); 4Milken Institute School of Public Health, George Washington University, 950 New Hampshire Avenue, Washington, DC 20052, USA; uriyoan@gwu.edu; 5Department of Family and Community Medicine, University of Illinois College of Medicine, Rockford, IL 61107, USA; ravneetk@uic.edu; 6Division of Kinesiology, Health and Sport Studies, College of Education, Wayne State University, Detroit, MI 48202, USA; rjankows@gmail.com (R.D.D.); bodebr@wayne.edu (B.B.); 7Urban Studies and Planning and Detroit Food Map Initiative, Wayne State University, Detroit, MI 48202, USA; alexbhill@wayne.edu; 8Departments of Health Policy Management & Nutrition, Harvard T.H. Chan School of Public Health, Harvard University, Boston, MA 02115, USA; jdeangelo@hsph.harvard.edu

**Keywords:** healthy food retail, food access, low-income, community food store, locally owned, case studies, qualitative

## Abstract

In the United States, low-income, underserved rural and urban settings experience poor access to healthy, affordable food. Introducing new food outlets in these locations has shown mixed results for improving healthy food consumption. The Healthy Community Stores Case Study Project (HCSCSP) explored an alternative strategy: supporting mission-driven, locally owned, healthy community food stores to improve healthy food access. The HCSCSP used a multiple case study approach, and conducted a cross-case analysis of seven urban healthy food stores across the United States. The main purpose of this commentary paper is to summarize the main practice strategies for stores as well as future directions for researchers and policy-makers based on results from the prior cross-case analyses. We organize these strategies using key concepts from the Retail Food Environment and Customer Interaction Model. Several key strategies for store success are presented including the use of non-traditional business models, focus on specific retail actors such as store champions and multiple vendor relationships, and a stores’ role in the broader community context, as well as the striking challenges faced across store locations. Further exploration of these store strategies and how they are implemented is needed, and may inform policies that can support these types of healthy retail sites and sustain their efforts in improving healthy food access in their communities.

## 1. Introduction

In the United States, low-income, underserved rural and urban areas are challenged by poor access to healthy, affordable food [1,2]. These areas tend to have low access to supermarkets and discount warehouse stores, whereas smaller food retailers (e.g., corner stores, convenience stores) are typically more common, providing easy access to food options, but offering an array of unhealthy options and high prices [2,3]. Policies and initiatives, such as the Healthy Food Financing Initiative (HFFI) [4], have supported healthy food retail projects over the last decade. The HFFI aims to improve healthy food access; for example, the introduction of supermarkets in low-income, low-access locations. However, such projects have shown mixed results in terms of improving food purchasing and consumption patterns [5]. These mixed results are due to many reasons, including low adoption of these new supermarkets as a regular shopping venue, and a lack of understanding of store- and community-level factors, such as pricing structures, marketing practices, and community buy-in that influence diet-related behaviors aside from access alone [5]. 

An alternative approach to understanding how to improve dietary quality is examining existing mission-driven, locally owned food stores who are already committed to improving healthy food access in their community. Such work would add to previous research examining grocery store accessibility and in-store interventions (e.g., pricing, placement) [6,7,8,9,10,11,12,13], by understanding which successful strategies and approaches may be scalable to other venues. It could also provide insight into the policy supports necessary to ensure that other food retailers are able to shift their missions to prioritize health and accessibility and identify key gaps in knowledge that should be addressed in future research.

The Healthy Community Stores Case Study Project (HCSCSP) was the first of its kind to explore this strategy, by delving into the structure and operations of seven mission-driven, locally owned food stores in low-income urban locations [14] using a multiple case study approach [15]. Additional details about the selection process of the seven stores, their distinct characteristics and contexts, and the study protocol have been previously published as part of this special issue [14]. Briefly, the seven stores were located in cities across the US mid-west and northeast who served low-to-middle income neighborhoods who were also predominantly immigrant or African American communities [14] The results of the cross-case analyses, including specific community engagement approaches and strategies used by stores to meet their healthy food accessibility missions, as well as barriers and facilitators, have already been described in other studies within this special issue [16,17,18,19]. The focus of this commentary is to draw on the insights from across the HCSCSP studies augmented with relevant literature from multiple fields (e.g., nutrition, food retail, business, and economics) to provide future practice, research, and policy implications that could promote the expansion of mission-driven work around healthy food accessibility across low-income communities.

We have organized our presentation of key insights using the Retail Food Environment and Customer Interaction Model [20], as others have previously done [21]. This model captures the retail food environment and customer relationships that lead to a range of societal challenges and helps to organize the complex and dynamic food environment into several, clearly defined components [20]. Below, we organize the insights drawn from across the HCSCSP studies by three key areas outlined in the model: business models; retail actors, and the larger macro-level contexts, including the socio-political landscape in which retailers and communities are embedded. Within each area, we describe evidence from the HCSCSP cases and literature, and then provide specific implications for stores, researchers, and policy-makers to consider.

## 2. Business Models

The business model component of the Retail Food Environment and Customer Interaction Model refers to the different business designs as well as ownership/financial models utilized which reflect the overall priorities and goals of a business [20]. Across the cases, we found four features of business models to be key, including: (1) store commitment to and investment in their community, (2) the type of ownership/financial model used, (3) the importance of hiring local staff, and (4) the implementation of needs-based and loyalty programs.

### 2.1. Commit to Community Engagement

Overall, a store’s commitment to community engagement was one of the most important factors in a business model [17,18]. This was even more important when socio-cultural differences were present among the store owners/managers and the community residents [19]. We found that each store approached their mission in different ways—in particular, how they engaged and interacted with their surrounding community [18]. All stores were operating in environments with complex and varied historical contexts, customer populations and demands, and financial considerations. That said, stores that took a broader role as a store serving their community demonstrated different priorities which helped shape strategies for improving healthy food access in their specific context. The following are specific practice implications for stores and future directions for researchers and policy-makers to consider as it relates to exploring ways to incorporate community engagement into a store’s business model.

*Store-Level Practice Implications*:
Learn about the historical contexts and resident dynamics in the community being served from the outset, and continue to learn over time. This can be achieved through partnering with community organizers and conducting formal or informal formative research that includes qualitative (interviews, focus groups) and quantitative data (community and/or customer surveys). Academic institutions could be key partners in conducting this type of research.Ensure frequent, bidirectional communication to and from the community. This may be accomplished through social media posting, hosting or attending community events, and customer suggestion boxes.Be clear about the store mission and its ability to achieve that mission to establish community trust. This may include producing annual reports or updates that are publicly available.Collaborate with key community stakeholders to invest in healthy food promotion activities (e.g., taste tests with a local school or healthy cooking classes with community members in store).*Future Research Directions*:
Explore how community engagement approaches vary across urban, suburban, and rural settings.Continue to explore the motivations among retailers that prioritize community engagement, and to inform scalability and sustainability.Explore how a community advisory board could be created and collaborated with to guide store-level policies and practices, as well as local-level policies.*Policy Implications*:
Consider incentivizing or financing stores that invest in community prosperity and health as part of their business model (e.g., through providing certificate or training programs tied to small financial incentives). Develop criteria for identifying stores who qualify to receive these types of incentives (related to store size, location, local ownership, mention of prioritizing healthy foods in the store mission, etc.).

### 2.2. Use New Ownership/Financial Models

Ownership/financial models in this project were diverse and included for-profit, non-profit, and cooperative models. As simply introducing new food outlets in areas with low food access has not been as successful as hoped, shifting gears from traditional business models, such as for-profit, to alternative business models such as non-profits or co-ops may ensure a sense of shared ownership in the community [18].

Non-traditional business models may also allow for more flexibility to stick to a healthy food-focused mission and offer healthy foods at an affordable price [17]. For example, some stores were able to create nutrition guidelines for provisioning healthy food, and received funding from non-traditional funding sources, such as foundations, fundraising, city grants, and individual donors. Non-profit stores in this project scored higher on the pricing component of the Healthy Food Availability for Healthy Eating Index, meaning that they were able to offer lower prices compared to other business model types [16].

The retail literature suggests that innovative business models also allow for flexibility and adaptability in response to the surrounding built and social environment through non-traditional practices [22,23]. Innovations to one or more of the main components of a business model (i.e., value proposition, value delivering, value creation, and value capturing) can determine the values a business creates, who receives the benefit of the values created, and how the business relates to consumers [22]. For example, a place-based business model makes use of location-specific resources (i.e., local producers, distributors) to create value [22]. This can ultimately lead to increased resilience and sustainability of the retailer, stronger customer relations, as well as the success of the local economy [22,23]. Moreover, non-economic goals of the retailer may reflect values shared by both the retailer and the consumer, and could thus be achieved through non-traditional business models such as co-ops, where the consumers are owners with multiple mechanisms to influence store decision-making [24].

*Store-Level Practice Implications*:
Consider a non-traditional business model that allows for shared ownership, leadership, and decision-making within the community. Allow for community input from the store’s inception including the selection of the location to developing the store’s healthy food mission and what healthy foods are sold in the store.Establish partnerships with local producers, gleaners, or other food donors to create other food sourcing options outside of traditional wholesalers and suppliers.Create nutrition guidelines for what can be provisioned and sold in the store. Involve the community in deciding what nutritional factors are most important to them in terms of what foods/nutrients to limit, and what foods/nutrients to encourage.*Future Research Directions*:
Develop insights into what type of business models current and future store owners would be willing to explore. Identify the barriers/facilitators to following each model to elucidate the best way to build capacity amongst local store owners to adopt non-traditional business models.Similarly, identify how policy-makers view different business models. Understanding what type of business models policy-makers would be more willing to support may inform future policies to financially support and incentivize new and existing stores.*Policy Implications*:
Streamline the process for becoming a non-profit grocery store, as well as the subsequent reporting, to be more feasible for store owners.Increase funding for technical assistance and capacity building among non-profit grocery store models to promote and protect sustainability, especially in places where food sovereignty is a public health goal.

### 2.3. Hire Local Staff

Business models also must consider who they hire and how this can add or detract from their mission. With retailers committed to the well-being of their communities, hiring staff from the community and empowering them to participate in decision-making can be essential for a business to thrive. Previous work has shown that businesses who hire local staff tend to experience a variety of benefits including lower recruitment costs, reduced turnover, and enhanced brand reputation [25]. Hiring locally also promotes inclusiveness, improves the local economy and helps build community capacity, bolstering the idea of the importance that stores recognize their broader role in the community [18,26,27].

*Store-Level Practice Implications*:
Aim to hire staff from the surrounding neighborhood and community. This may be achieved through local advertisements, word-of-mouth, attending community events to network, and social media postings.Establish partnerships with community organizations and existing programs to develop and/or participate in workforce training programs [26].*Future Research Directions*:
Explore key strategies that stores undertake to successfully hire and maintain local staff and document what is needed to inform strategies for other stores who would like to incorporate this practice into their business model.Examine how incorporating such a practice influences aspects of store success and sustainability, including community engagement, customer satisfaction, and foot traffic.*Policy Implications*:
Incentivize stores who hire local staff in low-income settings.Engage stores in local, state, and federal-level programs that provide workforce training opportunities for youth and young adults (e.g., AmeriCorps).

### 2.4. Implement Needs-Based or Loyalty Programs

Some stores in the project also employed a needs-based program or loyalty programs as part of their business approach. While the loyalty program incentivized repeat customers, the needs-based program made healthy food more accessible to groups with limited resources [17]. Specifically, the needs-based program helps to level the playing field by having those who identify as ‘needs-based’ pay a lower price at the check-out every time. The use of innovative, non-traditional strategies, such as the needs-based program as well as the ability to accept donations and collaborate with innovative suppliers, can help equitably address both the affordability aspect for consumers as well as offset the costs for retailers to ensure financial viability.

*Store-Level Practice Implications*:
Offer a loyalty program or needs-based program for repeat and/or lower-income customers to increase the affordability of the healthy foods being sold, and build customer loyalty. A needs-based program provides a discount at every check-out for those who qualify based on self-reported need or income (in this project, customers received a 10% discount).Offer discounts and promotions to be competitive with surrounding stores.Conduct competitive price analyses regularly.*Future Research Directions*:
Further examination of different customer programs/incentives, how they operate and what their impact is on food access and healthy food purchase is needed, especially across business model type.Test the impact of implementing loyalty and needs-based programs on customer satisfaction, foot traffic, and purchasing patterns of healthy foods.*Policy Implications*:
Incentivize pricing structures that further enable access to low-income shoppers, such as a needs-based program.

## 3. Retail Actors

Retail food actors are the people that work within the retail food environment at different levels across the food supply chain, such as managers, distributors, and sales representatives [20]. In this multiple case study project, we found two key actors to be essential to mission-driven, locally owned food stores: store champions and vendors.

### 3.1. Identify and Work with Store Champions

Store champions were present in all seven stores, and were typically a higher-level staff member with great passion and commitment to the store as well as to its healthy food mission [17]. The importance of a store champion, or local community business owners/staff, on influencing community health values and being agents of health promotion has been demonstrated in the context of supermarkets. For instance, Tesco in the United Kingdom has over 300 fully funded community champions that act as the link between the store and the community [28], as well as other business settings [17,29,30,31,32,33].

*Store-Level Practice Implications*:
If the store does not have a local store champion (i.e., it is not locally owned), consider hiring a manager from the surrounding community who is passionate about improving food access and health in the community [28]. This person should be involved in store decision-making and trained on other aspects of store management and operations; however, the general passion and commitment to this work cannot necessarily be learned.*Future Research Directions*:
More research is needed to explore the role of a store champion, identifying strategies for engaging active community members who could become a store champion in stores that do not already have one as well as understanding the impact that gaining a store champion can have on store success.

### 3.2. Diversify Vendors

Another set of retail actors identified as key to store success in this project were product vendors. Many of the stores who utilized multiple vendors versus one or a few were able to offer lower prices to their customers [17]. Product vendors are key to business success, and not having adequate vendor relations has been identified as a barrier to entry into the market for new stores, given vendors may already have strong relationships with existing stores and/or larger chains [34,35].

The retail literature also suggests that the supply chain for unhealthy foods (i.e., salty and sweet snacks, sugar-sweetened beverages) in small store settings is more robust than that of healthier options (i.e., fresh produce); unhealthier products can often be delivered to the store (whereas it is often not possible, or more expensive to have healthier items delivered), and vendors may offer incentives for purchasing these products [36,37,38]. Larger product vendors often have minimum order requirements which can be challenging to manage for small stores, especially for healthy perishable items that can spoil and detract from the business bottom line [36,37,39]. Stores can work together in collaboration with vendors to obtain multiple vendor options (traditional and non-traditional, such as local producers, gleaners, donors) and improve the supply chain of healthy foods to smaller stores, which could be supported by policies that incentivize these relationships. Perhaps focusing on local store–vendor relationships would lend itself to collaboration and be of interest to local policy-makers interested in improving healthy food access in their communities [39], as well as at the federal level where input is currently being solicited for better supporting small, mid-sized, and independent processors [40]. In addition, having multiple avenues for sourcing food may increase a store’s resiliency during supply chain interruptions such as those experienced during the COVID-19 pandemic [41].

*Store-Level Practice Implications*:
Engage multiple vendors by dedicating staff time to explore local, regional, and national purchasing opportunities.Consider working with local vendors to further increase community relations and support the local economy.Consider working with other local small retailers to coordinate collective purchasing that creates economies of scale.*Future Research Directions*:
Further explore how existing stores identify and maintain relationships with multiple vendors to understand the process and provide guidance for stores who would like to increase the number of vendors they utilize. Of interest may be assessment of stores who already practice collective purchasing with partner stores, and testing this strategy more broadly. Development of digital strategies and platforms with user-friendly features that facilitate these relationships may be beneficial.*Policy Implications*:
Consider providing tax incentives to vendors who distribute healthier food items in smaller batches to small stores, or who are working to build capacity to do so.Incentivize local food distribution hubs that can support community store fresh produce orders.Support initiatives that aim to localize the food supply chain and thus facilitate the establishment of new local vendor–retailor relationships.

## 4. State, Tribal, National, and Global Context

The state, tribal, national, and global context of the Retail Food Environment and Customer Interaction Model refers to macro-level factors and contexts, such as policies, that can influence all aspects of the retail food environment, including the surrounding community, retail actors, and relationships [20]. In this section, we reflect on the various contexts that impacted stores in our study and the policy solutions that could be implemented to help address and mitigate these larger contextual forces. Specifically, we discuss the impacts from the COVID-19 pandemic and unintentional constraints placed on small stores from federal food policies.

Unmitigated structural racism has resulted in unequal access to healthy affordable food (among other critical resources) particularly in Black and Latinx communities [42]. One way to begin to take action to address these inequities in food access and the consequential negative health outcomes is to support smaller, locally owned, mission-driven food retailers like the ones included in this case study project, as opposed to larger chains. One store received city funding to support the store in stocking and selling of healthier options, highlighting the importance of supporting the “supply-side” of food retail, not just the demand side, which is often the focus of federal-level policies related to food assistance programs. Strategies such as federal-level funding or state and local tax breaks for stores who prioritize making healthy foods affordable may incentivize new and existing stores to focus on health.

### 4.1. Impacts of the COVID-19 Pandemic

Many food retail stores suffered during the COVID-19 pandemic due to overall decreased foot traffic [34], supply chain shortages [43], increased prices [43,44], and the inability, particularly for small stores, to offer online shopping or delivery options [45,46]. Small stores such as the ones included in this case study were forced to look for other funding opportunities, and in one case, the store closed its doors, while the Paycheck Protection Program mostly supported larger companies. Although the business strategies outlined above (i.e., use of multiple vendors, flexibility in where food and funding can come from) cannot completely protect a business from the impact of an event like the COVID-19 pandemic, it may help increase resiliency during these times of hardship by having many supporting partners and avenues for continuing to serve food in their communities [41].

*Future Directions for Research*:
Explore variation in COVID-19 adaptations by urban, suburban, and rural settings.Based on lessons learned from store adaptations during the COVID-19 pandemic, develop strategies and protocols for store emergency preparedness in collaboration with store staff.*Policy Implications*:
Prioritize smaller stores with fewer staff and resources in emergency relief acts in future national or global crises.Provide ample opportunities for technical assistance and other supports targeted to smaller, locally owned, mission-driven stores who have less capacity to navigate bureaucratic forms and applications.

### 4.2. Federal Food Policy Constraints

At the federal level, small, locally owned food stores should be considered in future policies related to the Supplemental Nutrition Assistance Program (SNAP) and the Special Supplemental Nutrition Program for Women, Infants, and Children (WIC). Expansion of guidelines to allow smaller retailers to feasibly offer and maintain acceptance of food assistance benefits may improve food security in the communities they serve. For example, expanding the WIC eligible food product list to include alternative healthy food options, outside of conventional products, may encourage smaller stores to participate. In one store, a large barrier to WIC participation was the constraints of the food product list. The store had to collaborate with other retailers in their city to be able to order the specific conventional products required by WIC and offer them in the store, as their preferred, environmentally sustainable products did not qualify. Ultimately, consideration for these smaller stores with fewer resources but significant mission-driven work is needed in future policy formation as they are vital to serving the community. 

*Future Directions for Research*:
Simulate or test the feasibility of the potential policies and strategies described above, using tools such as casual loop diagrams and system dynamics modeling, to assess their potential impacts, as has been done in other studies [12,47,48,49,50].Explore the acceptability and perceived feasibility of such policies and strategies among store owners and policy-makers, and understand their views on how these processes could be streamlined and made more accessible.*Policy Implications*:
Expand federal nutrition program guidelines to make participating as a WIC and/or SNAP vendor more accessible to smaller, locally owned, mission-driven stores with fewer resources.Provide technical assistance centers for SNAP-authorized retailers that could provide best practices, resources on sourcing, and stocking healthy foods.Expand the Gus Schumacher Nutrition Incentive Program (GusNIP) to community stores to financially incentivize fruit and vegetable purchases made with SNAP benefits at these stores.Collaborate with Healthy Food Incentive Programs at the state level, such as with Women, Infants, Children (WIC) and/or Supplemental Nutrition Assistance Programs (SNAP) so that loyalty programs and discounts/coupons are connected to consumer accounts with WIC and/or SNAP.

## 5. Conclusions

The Healthy Community Stores Case Study Project was novel in its attempt to deeply explore how mission-driven, locally owned food stores who prioritize healthy food navigate their complex local food systems in order to survive and thrive. This project brought to light several key strategies for store success as described here, as well as apparent trials and tribulations faced across store locations during standard operations along with the events of 2020 with the COVID-19 pandemic. Further exploration of these store strategies and how they are operationalized across urban, suburban, and rural locations is needed in order to test them in other stores, as well as to inform policies that support these types of healthy retail sites and sustain their efforts in low-income, low-healthy food access communities.

## Data Availability

Not applicable.

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
