# Peer review of "A Commentary on the Healthy Community Stores Case Study Project: Implications for Retailers, Policy, and Future Research"

_ijerph, 2022, doi:10.3390/ijerph19148824_

Round 1

Reviewer 1 Report

The Health Community Stores Case Studies Project article delivers what it promises. Though an examination of a corpus of material focusing on mission-driven locally-owned food stores that are committed to improving healthy food access in their communities, the authors provide evidence-based implications for the retailers themselves and for policymakers. Future research questions are also identified. The authors specifically examine business models, retail actors, and the broader state, tribal, national and global contexts for this mission-driven economic activity.

The paper is well-organized and clearly presented, allowing readers with different interest areas to identify content that would be particularly relevant to their interests and needs. 

I would recommend further elaboration of the specific contexts of the 7 stores. The introduction refers generally to "low-income, underserved rural and urban areas," but no further information is provided to the reader. Are the 7 sites in fact urban and rural? How many cities or rural areas are represented? Which low-income underserved communities are in fact served by the 7 stores in the study? If possible, including these details would enhance the readers' understanding of the social context of the study and help the authors address the implicit question of how these findings might be applied to other contexts and scaled.

Reviewer 2 Report

The topic developed in the article is interesting, but some implementations are needed in my opinion. The first point to develop concerns the model you apply. A part of the same authors presented this model in the 2020 article. It would be interesting to analyze the process of choosing the model, compared to other methodologies for the study of similar issues.

Has this model been used by other scholars?

If so, reference literature should be included.

Another weakness is the champion. 7 points of sale were identified. First, it should be included in the title and then in the analysis that your research is an exploratory investigation. The sample is limited. What were the criteria for choosing these stores?

The bibliographic research is discreet. In my opinion, a thorough search should be conducted on the main scientific search engines to look for references that can enrich the bibliography since less than 30% has been published in the last five years.

Reviewer 3 Report

The study presents important insights from the Healthy Community Stores Case Studies Project (HCSCSP) which aims to improve healthy food access for disadvantaged neighborhoods. This paper draws from the Retail Food Environment and Customer Interaction Model to devise the most effective strategies for grocery stores and directions for future research and policy.

The paper is interesting and adds to our understanding of the problems with healthy food delivery to disadvantaged communities. However, the information that the authors provide about the HCSCSP itself is not sufficient. In the abstract, the authors say that the program “used a multiple case study approach and conducted a cross-case analysis of seven urban healthy food stores across the United States.” What were these ‘healthy stores’? Where were they located? What were the results of the ‘cross-case analysis’?

Moreover, it is not clear whether the results of this program are generalizeable to other urban locales in the United States? Are the strategies that the authors promote applicable to rural grocery stores? It is known that rural areas with declining populations may be specifically at risk for low access to healthy foods.

Minor issues:

Abstract: “future directions for researchers and policymakers”: Use ‘policy makers’ to be consistent with the text.

Page 2, middle of the page: “The case study approach allowed for preservation”, insert ‘the’ before ‘preservation’.

Page 3, last paragraph: Change ‘heathy’ to ‘healthy’.

Page 9, top of the page: Should be ‘at the state level’.

Round 2

Reviewer 2 Report

In my opinion, the article can be published

Reviewer 3 Report

All relevant reviewer comments have been addressed; consequently the manuscript can be published in its current form.